

**A High-Accuracy Rainfall Dataset by Merging Multi-Satellites**
**and Dense Gauges over Southern Tibetan Plateau for 2014-2019**
**Warm Seasons**
Kunbiao Li[1], Fuqiang Tian[1], Mohd Yawar Ali Khan[2], Ran Xu[1], Zhihua He[3], Long
Yang[4], Hui Lu[5], Yingzhao Ma[6]
[1] Department of Hydraulic Engineering, Tsinghua University, Beijing, China
[2] Department of Hydrogeology, King Abdul-Aziz University, Jeddah, Saudi Arabia
[3] Centre for Hydrology, University of Saskatchewan, Saskatoon, SK S7N 5C8,
Canada
[4] School of Geography and Ocean Science, Nanjing University, Nanjing, China
[5] Department of Earth System Science, Tsinghua University, Beijing, China
[6] Department of Civil & Environment Engineering, Colorado State University, FT
COLLINS, CO, USA
**Correspondence:** Fuqiang Tian (tianfq@mail.tsinghua.edu.cn)





**Abstract**

Tibetan Plateau (TP) is well known as the Asia's water tower from where many large
rivers originate. However, due to complex spatial variability of climate and topography,
there is still a lack of high-quality rainfall dataset for hydrological modelling and flood
prediction. This study, therefore, aims to establish a high-accuracy daily rainfall product
through merging rainfall estimates from three satellites, i.e., GPM-IMERG, GSMaP,
and CMORPH, based on the likelihood measurements of a high-density rainfall gauge
network. The new merged daily rainfall dataset with a spatial resolution of 0.1°, focuses
on warm seasons (June 10th - October 31st) from 2014 to 2019. Statistical evaluation
indicated that the new dataset outperforms the raw satellite estimates, especially in
terms of rainfall accumulation and the detection of ground-based rainfall events.
Hydrological evaluation in the Yarlung Zangbo River Basin demonstrated high
performance of the merged rainfall dataset in providing accurate and robust forcings
for streamflow simulations. The new rainfall dataset additionally shows superiority to
several other products of similar types, including MSWEP and CHIRPS. This new
rainfall dataset is publicly accessible at https://doi.org/10.11888/Hydro.tpdc.271303
(Li et al.,2021).

**1. Introduction**


Precipitation, linking atmospheric and hydrological processes, serves as a crucial
component of the water cycle (Eltahir & Bras, 1996; Trenberth et al., 2003). Gridded
precipitation datasets become more and more popular with the advent of satellite
precipitation measurement. Most famous satellite gridded precipitation datasets include
Tropical Rainfall Measuring Mission (TRMM) (Huffman et al., 2007) and its successor
the Integrated Multi-satellite Retrievals for Global Precipitation Measurement mission
(GPM-IMERG) (Hou et al., 2014), the Global Satellite Mapping of Precipitation



(GSMaP) (Ushio et al., 2009), the Climate Prediction Centre (CPC) MORPHing
technique (CMORPH) (Joyce et al., 2004), etc. These products have been successfully
applied in various hydrometeorological studies and water resources management
practices (Kidd, C., & Levizzani, V., 2011; Jiang et al., 2012; Tong et al., 2014; Yang et
al., 2015; Sun et al., 2016; Wang et al., 2017).
However, all existing precipitation datasets show insufficient accuracy in high
mountainous regions (Yilmaz et al., 2016; Derin et al., 2018; Derin et al., 2019;
Anagnostou & Zhang, 2019), which hinders our understanding of climate and
hydrological processes over these areas. This can be attributed to the complex physical
nature of electromagnetic transmission and precipitation forming processes (Hong et
al., 2007; Bitew & Gebremichael 2010; Dinku et al., 2010), and harsh environments in
high mountains that lead to very limited deployment of in-situ rain gauges with
insufficient representation of ground observations for training satellite-based
precipitation retrieval algorithms. For instance, the Tibetan Plateau (TP) as the roof of
the world is surrounded by imposing mountain ranges with an average elevation
exceeding 4000 m. It generates several large rivers in Asia and provides invaluable
freshwater resources for more than 1.4 billion people living downstream (Immerzeel et
al., 2010). However, this vast plateau has very limited number of precipitation gauges
across its 2.5 million $km^2$ area. The precipitation gauge network operated by China
Meteorological Agency (CMA) contains only 86 gauges over the entire TP (Figure 1).
These gauges are essential to correct satellite precipitation datasets. For example, GPM-
IMERG 'Final' Run dataset uses Global Precipitation Climatology Centre (GPCC)
database, GSMap_Gauge and CMORPH use NOAA Climate Prediction Centre (CPC)
database. Although both GPCC and CPC databases received data through Global
Telecommunication System (GTS), only part of the above-mentioned gauges in TP
were utilized (Xie et al., 2007; Becker 2013). Previous evaluations over the TP
indicated that most products present dependence on topography to varying degrees, and
products adjusted by gauge observations shows better performance than satellite-only



products (Gao et al.,2013; Lu et al., 2018). Therefore, a better spatial coverage of rain
gauges is critical to correct satellite products in high mountains.

In 2014, the Ministry of Water Resources of China (MWR) launched the flash

flood monitoring and alarming campaign. A large number of rain gauges is now
accessible over the TP, especially in the southern TP. There are 440 new rain gauges,
and are available since 2014, independent of the existing CMA precipitation gauge
network (Figure 1). These gauges provide measurements of precipitation in liquid phase
(i.e., rainfall) at event time scale. A couple of recent studies have demonstrated the
utility of this rain gauge network (Xu et al., 2017; He et al., 2017; Tian et al., 2018;
Wang et al., 2020). For instance, Xu et al. (2017) evaluated the performance of TRMM
and GPM and the dependence on topography and rainfall intensity based on the network.
Their results demonstrated that the data quality of this dense gauge network is strictly
controlled, serving as the currently highest gauge dense for satellite product evaluation
on TP. Wang et al. (2020) used the gauge data to validate their reproduced precipitation
dataset. However, there is not a merging product that assimilate the observations from
this dense rain gauge network. This is apparently a unique opportunity to improve the
performance of existing satellite-based precipitation datasets for its highest density and
quality.

This study aims to provide a high-accuracy rainfall dataset by merging all

available ground gauges and three good-quality satellite precipitation datasets over the
southern TP for the warm seasons (June 10$^{th}$ - October 31$^{st}$) from 2014 to 2019. The
remainder of this paper is organized as follows: Section 2 describes the study area and
the source data. Section 3 provides details of the data merging method and the methods
adopted to evaluate the quality of dataset. Results are presented in Section 4. The data
availability and summary are provided in Section 5 and Section 6, respectively.

## 2. Study Area and Source Data

### 2.1. Southern Tibetan Plateau

Tibetan Plateau, known as the Asian water tower, borders India, Myanmar, Bhutan and Nepal to the south and Pakistan to the west. Various climate systems affect the plateau, including westerly winds in winter and the Indian monsoon in summer (Yao et al., 2012). Many Asian large rivers originate from this vast area, including the Yellow River, the Yangtze River, the Yarlung Zangbo River (YZR), Jinsha River (JR), Lancang River (LR), Salwen River (SR), Irrawaddy River (IR), Ganges River (GR), and Indus River (IDR). This study is focused on the southern part of TP (Figure 1), including the upper YZR Basin (YZRB) as a major basin.

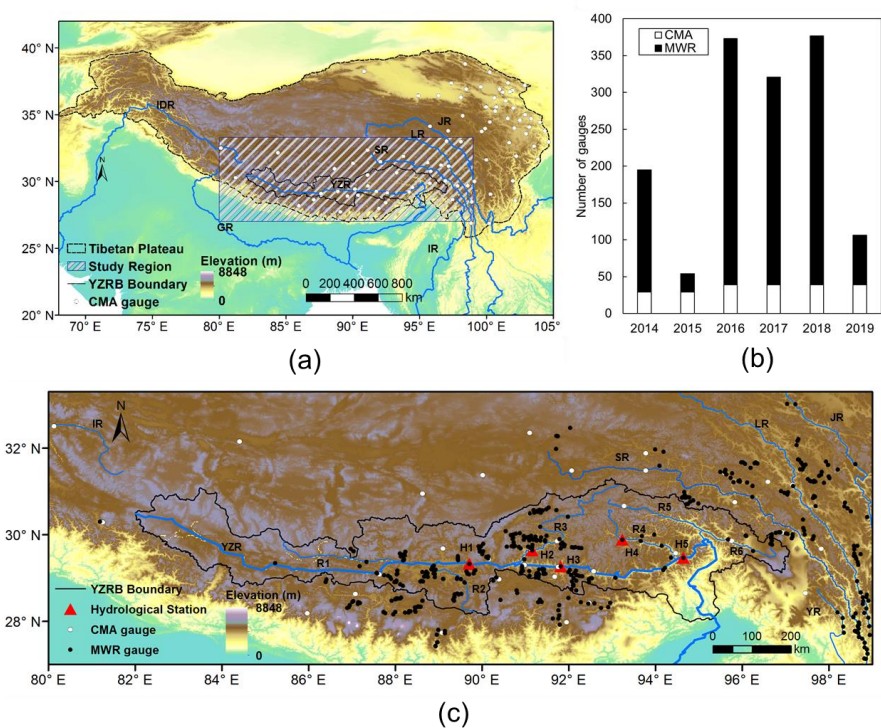

Figure 1. (a) The location and topography of the TP and the spatial distributions of CMA gauges.

(b) Numbers of ground gauges installed by CMA and MWR in southern TP during 2014-2019, (c)





108 Locations of CMA and MWR rain gauges and main hydrological stations in southern TP. The names

109 of hydrological stations are labelled as H1-Yangcun, H2-Lhasa, H3-Nugesha, H4-Gongbujiangda,

110 H5-Nuxia. The names of tributary rivers are labelled as R1-Duoxiong Zangbo, R2-Nianchu River,

111 R3-Lhasa River, R4-Niyang River, R5-Yigong Zangbo, R6-Parlung Zangbo.

112 **2.2. Ground gauged rainfall**

113 We combined two rain gauge networks managed by MWR and CMA to obtain a

114 high-quality ground reference dataset up to date. The number of rain gauge is presented

115 in Figure 1b, and varies across different years. The spatial distribution of all gauges is

116 presented in Figure 1c. The gauges are mainly located in the middle reaches of YZRB

117 and the east part of the study area. Despite the high density, we can see these rain gauges

118 are not evenly distributed across the space. This makes satellite rainfall products over

119 varying altitudes and aspects important. Daily rainfall observations during the warm

120 seasons of 2014-2019 were accumulated from the original event scale measurements.

121 Total number of the CMA and MWR gauges ranges from 53 in 2015 to 377 in 2018,

122 forming the densest rain gauge network up till now.

123 The CMA gauge data has been widely demonstrated as reliable and accurate in

124 previous studies (Zhai et al., 2005; Su et al., 2020; He et al., 2020). Gauge data used in

125 this study has been manufactured under strict quality control procedures, including (1)

126 internal consistency check, (2) extreme values check (0~85mm/h), and (3) spatial

127 consistency check (Ren et al., 2010). Rain gauges with erroneous values (e.g.

128 enormously large values) were discarded from the entire records.

129 **2.3. Satellite Precipitation Datasets**

130 Three satellite precipitation products were chosen for the data merging procedure

131 (Lu et al., 2019; Derin et al., 2019; Tang et al., 2020), including GPM-IMERG 'Final'

132 run (here after referred to as IMERG) from the National Aeronautics and Space

133 Administration (NASA) (https://disc.gsfc.nasa.gov/), the GSMaP_Gauge (here after



referred to as GSMaP) from Japan Aerospace Exploration Agency (JAXA)
(http://sharaku.eorc.jaxa.jp) and the CMORPH v1.0 from NOAA CPC
(ftp://ftp.cpc.ncep.noaa.gov/precip/CMORPH_V1.0/). Spatial resolutions and temporal
frequency of the satellite datasets are listed in Table 2. To be consistent, IMERG and
GSMaP data were accumulated to daily scale (08:00-08:00 of local time, i.e. UTC+8)
and CMORPH was bilinearly interpolated to the grid resolution of 0.1°.
The merged dataset was further compared with two popular merged rainfall
datasets of Climate Hazards Group InfraRed Precipitation with Stations (CHIRPS)
(Funk et al., 2015) and Multi-Source Weighted-Ensemble Precipitation (MSWEP)
(Beck et al., 2019). CHIRPS was originated by merging CHPClim, thermal infrared,
TRMM3B42, NOAA CFSv2 precipitation data, and ground observation precipitation
data. MSWEP was merged from multiple datasets including CPC, GPCC, CMORPH,
GSMaP-MVK, GPM-IMERG, ERA5, and JRA-55. CHIRPS and MSWEP showed
great potentials in rainfall estimates in previous studies (Liu et al., 2019).
Table 2. Multiple satellite precipitation datasets used in this study.

| Datasets | Resolution | Frequency | Source | Reference |
|---|---|---|---|---|
| GPM IMERG | 0.1 °x 0.1 ° | 0.5 hourly | NASA | (Hou et al., 2014) |
| GSMaP_Gauge | 0.1 °x 0.1 ° | 1 hourly | JAXA | (Ushio et al., 2009) |
| CMORPH v1.0 | 0.25 °x 0.25 ° | daily | CPC | (Joyce et al., 2004) |
| CHIRPS v2.0 | 0.25 °x 0.25 ° | daily | USGS and CHC | (Funk et al., 2015) |
| MSWEP v2 | 0.1 °x 0.1 ° | 3 hourly | - | (Beck et al., 2019) |

**3. Methodology**
We used the Dynamic Bayesian Model Averaging (DBMA) method (Ma et al.,
2017) to merge the satellite datasets with in-situ rain gauges. To evaluate the quality of
the new dataset, we carried out statistical and hydrological evaluations and comparisons
with CHIRPS and MSWEP in southern TP.

### 3.1. Dynamic Bayesian Model Averaging method


The Dynamic Bayesian Model Averaging (DBMA) method developed by Ma et
al. (2018) was utilized in this work. A flow chart of the merging method is shown in
Figure 2. In the first step, a training dataset was formed by selecting samples from the
ground gauged data and three original satellite datasets. The training period was set as
40 days. Increasing the length of the training period did not lead to obvious
improvement of the merging method (Ma et al., 2018). In the second step, the training
dataset was transformed by the Box-Cox Gaussian distribution, and the optimal weights
for each of the original satellite datasets on a specific grid where a ground gauge is
located on each training day were estimated by a logarithmic likelihood equation and
the optimal expectation algorithm. In the third step, an ordinary Kriging interpolation
method was applied to spatially interpolate the daily weights onto grids with no gauges.
Finally, posterior spatiotemporal weights were used to obtain the final merged rainfall
dataset. The DBMA-merged data has been proved in Ma et al. (2017) to outperform
original satellite data during 2007-2012 over TP.

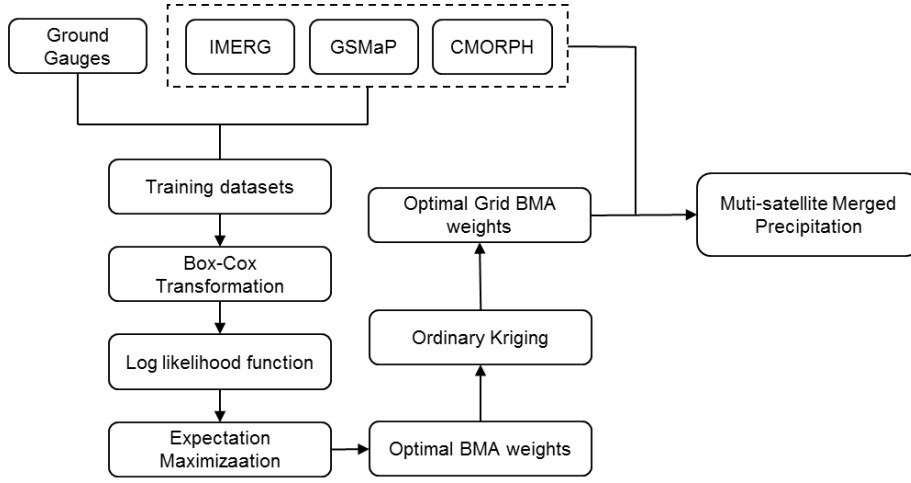


Figure 2. Flowchart of the DBMA merging method (adapted from Ma et al., 2018).
For statistical evaluation of the merged data against ground gauges, around 85%
of the gauges were randomly selected to form a training gauge set for the merging



approach in each year during 2014-2019, and the remaining 15% were used for test.
Table 1 lists the numbers of training and test gauges in each of the warm seasons. The
spatial distributions of gauges in each year are presented in Figure S1. Data from all
gauges were involved in the training procedure of the final released version of the
merged data.

Table 1. Number of rain gauges for training and test in 2014-2019.

| Year | Total number of rain gauges | Number of training gauges | Number of test gauges |
|---|---|---|---|
| 2014 | 195 | 166 | 29 |
| 2015 | 54 | 46 | 8 |
| 2016 | 373 | 317 | 56 |
| 2017 | 321 | 273 | 48 |
| 2018 | 377 | 320 | 57 |
| 2019 | 106 | 90 | 16 |

**3.2. Statistical Evaluation**
Performance of the multiple datasets were statistically evaluated by comparing
with ground observations on the corresponding statelite grids. Relative bias (RB) and
normalized root mean square error (RMSE) were adopted to measure the amount
difference between the gridded rainfall and the gauged rainfall. Correlation Coefficient
(CC) was used to evaluate the consistency between satellite estimates and gauge
observations. The skill of rainfall data on detecting rainfall occurrence (rainfall events
higher than zero) was evaluated through a set of metrics (similarly to Wilks, 2006): i.e.
the probability of detection (POD) assessing how good the multiple rainfall datasets are
at detecting the occurrence of rainfall, false alarm ratio (FAR) measuring how often the
gridded rainfall datasets detect rainfall when there actually is not rainfall, and critical
success index (CSI) measuring the ratio of rainfall events that are correctly detected by
the gridded datasets to the total number of observed or detected events. Equations for
the above metrics are shown in Table 2.

Table 2. Statistical indices that were used to assess the performance of the gridded rainfall





datasets.

| Statistical Indicators | Equation | Optimal Value | Equation number |
|---|---|---|---|
| Relative Bias (RB) | $Bias = \dfrac{\sum_{i=1}^{n}(S_i - G_i)}{\sum_{i=1}^{n} G_i}$ | 0 | (1) |
| Correlation Coefficient (CC) | $CC = \dfrac{[\sum_{i=1}^{n}(S_i - \bar{S}) \cdot (G_i - \bar{G})]^2}{\sum_{i=1}^{n}(S_i - \bar{S})^2 \cdot \sum_{i=1}^{n}(G_i - \bar{G})^2}$ | 1 | (2) |
| Root Mean Square Error (RMSE) | $RMSE = \sqrt{\dfrac{1}{n}\sum_{i=1}^{n}(S_i - G_i)^2}$ | 0 | (3) |
| Probability of Detection (POD) | $POD = \dfrac{a}{a + c}$ | 1 | (4) |
| False Alarm Ratio (FAR) | $FAR = \dfrac{b}{a + b}$ | 0 | (5) |
| Critical success index (CSI) | $CSI = \dfrac{a}{a + b + c}$ | 1 | (6) |

For the equations listed in Table 2, n is the total number of gridded product data

and gauge observation data; i is the $i^{th}$ of satellite product data and gauge
observation data; $G_i$ means gauge observation and $\bar{G}$ is the average of gauge
observation. $S_i$ and $\bar{S}$ are gridded estimates and their average, respectively. a
represents hit (i.e., event was detected to occur and observed to occur), b represents
false alarm (i.e., event was detected to occur but not observed to occur), and c
represents miss (i.e., event was not detected to occur but observed to occur).

Triple Collocation (TC) technique provides a platform for quantifying the root

mean square errors of three products that estimate the same geophysical variable
(Stoffelen, 1998). Roebeling et al. (2012) successfully applied the TC technique to
estimate errors of three rainfall products across Europe. An extended Triple Collocation
(ETC) introduced in Kaighin et al. (2014), which is able to estimate errors and
correlation coefficients with respect to an unknown target was used in this study to
compare the performance of the DBMA-merged data and two previous merged datasets
of CHIRPS and MSWEP.



### 3.3. Hydrological Evaluation

In addition to the statistical assessments against rain gauges, hydrological assessment was used as a tool to test the performance of merged rainfall datasets on forcing hydrological modelling in the study area (similarly see Yong et al, 2012; Xue et al, 2013; Yong et al, 2014; Li et al, 2014). In this section, a semi-distributed hydrological model developed by Tian (2006), namely Tsinghua Hydrological Model based on Representative Elementary Watershed (THREW), was adopted for the hydrological assessment of rainfall datasets in the YZRB. YZRB has a drainage area of approximately 240,480 $km^2$ within China's boarder. The basin elevation ranges from 143 to 7,261 m, with an average of around 4,600 m. YZR is one of the most important transboundary rivers in South Asia and the highest river in the world, which is characterized by a dynamic fluvial regime with exceptional physiographic setting spreading along the eastern Himalayan region (Goswami, 1985). Due to complex terrain and strongly varying elevation, the YZRB is under control of a variety of climate systems, such as the semi-arid plateau climate prevailing in the upper and middle reaches, and the mountainous subtropical and tropical climates prevailing in the lower reaches. In the cold upper reaches, the mean annual rainfall is less than 300 mm. In the warm middle reaches, the mean annual rainfall falls between 300 mm and 600 mm.

The whole basin area above the Nuxia hydrological station was divided into 63 Representative Elementary Watersheds (REWs). Model parameters were calibrated by daily discharges measured at the Nuxia station. The calibration period is scheduled to run in the warm seasons from June 10th to October 31st in 2014- 2017, encompassing a period length of 576 days. The validation period includes two warm seasons in 2018 and 2019 with a total duration of 288 days. Descriptions of the calibrated model parameters can be found in Table 3. An automatic algorithm pySOT developed by D. Eriksson et al (2019) was used to optimize the parameter values based on an objective-function of *NSE* (Nash and Sutcliffe, 1970) in Eq. 7. To conduct a continuous





hydrological simulation in the study period, the datasets of daily grid-based

precipitation over China (Zhao et al., 2014) were used as model inputs in the non-warm

seasons when merged rainfall is not available.

Table 3. Calibrated parameters of the THREW model.

| Symbol | Description | Unit | Value Range |
|---|---|---|---|
| $kv$ | Fraction of potential transpiration rate over potential evaporation | - | 0.001-0.8 |
| $n^t$ | Manning roughness coefficient for hillslope | - | 0.0001-0.2 |
| $GaIFL$ | Spatial heterogeneous coefficient for infiltration capacity | - | 0.0001-0.7 |
| $GaEFL$ | Spatial heterogeneous coefficient for exfiltration capacity | - | 0.0001-0.7 |
| $GaETL$ | Spatial heterogeneous coefficient for evapotranspiration capacity | - | 0.0001-0.7 |
| $WM$ | Tensor water storage capacity | cm | 0.1-10 |
| $B$ | Shape coefficient to calculate the saturation excess runoff area | - | 0.01-1 |
| $Gaus$ | Coefficient representing spatial heterogeneity of exchange term between t-zone and r-zone | - | 0.001-10 |
| $KKA$ | Exponential coefficient to calculate subsurface flow | - | 0.01-6 |
| $KKD$ | Linear coefficient to calculate subsurface flow | - | 0.001-0.5 |
| $MM$ | Snow melting degree-day factor | mm/day | 0.001-10 |
| $MMG$ | Ice melting degree-day factor | mm/day | 0.001-10 |
| $C1+C2$ | Muskingum parameter | - | 0.0001-1 |
| $C1/(C1+C2)$ | Muskingum parameter | - | 0.0001-1 |

$$NSE = 1 - \frac{\sum_{n=1}^{N}(Q_{obs}^n - Q_{sim}^n)^2}{\sum_{n=1}^{N}(Q_{obs}^n - \bar{Q}_{obs})^2} \qquad (7)$$

where, $N$ is the total number of days in the evaluation period, $Q_{obs}^n$ and $Q_{sim}^n$

represent the observed and simulated runoff on the $n^{\text{th}}$ day, respectively. $\bar{Q}_{obs}$

represents the average of observed runoff in the evaluation period.



## 4. Results and Discussions

### 4.1. Spatiotemporal Patterns

Based on the merging method, a new daily rainfall dataset with spatial resolution of 0.1°×0.1° in the warm seasons from June 10[th] to October 31[st] (144 days in each year) in 2014-2019 (864 days in six years) was generated. Figure 3 presents the spatial pattern of the mean rainfall over the six warm seasons of the merged data in southern TP. It is shown that extremely high summer rainfall centres concentrate in the south-eastern and south-western of the study area where is known as a world-famous heavy rainfall centre (see Biskop et al., 2015; Bookhagen & Burbank, 2006; Kumar et al., 2010).

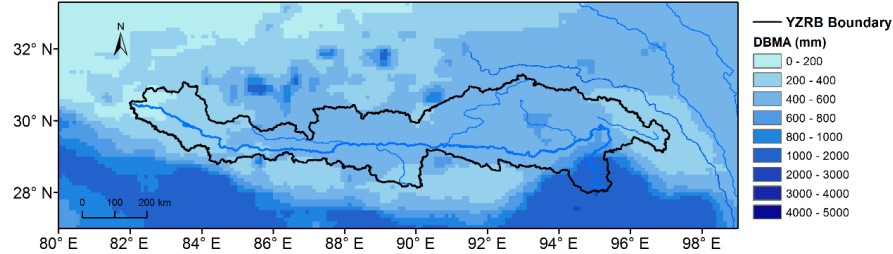

Figure 3. Spatial pattern of mean rainfall over six warm seasons in 2014-2019 of the DBMA-merged data in southern TP.

In addition, Figure 4 compares the time series of average daily weight and rainfall over the YZRB basin derived from the DBMA-merged data and the original satellite datasets. As expected, the DBMA-merged daily rainfall in general fall in the envelope ranges of the three satellite datasets. Merged data is closer to CMORPH in June, September and October, while showing equal closeness to all the three source satellite data in July and August. It indicates that CMORPH is closer to the in-situ gauges than IMERG at basin scale when rainfall value is small, especially for light rainfall events smaller than 2 mm, but this difference tends to be small for heavy rainfall events.



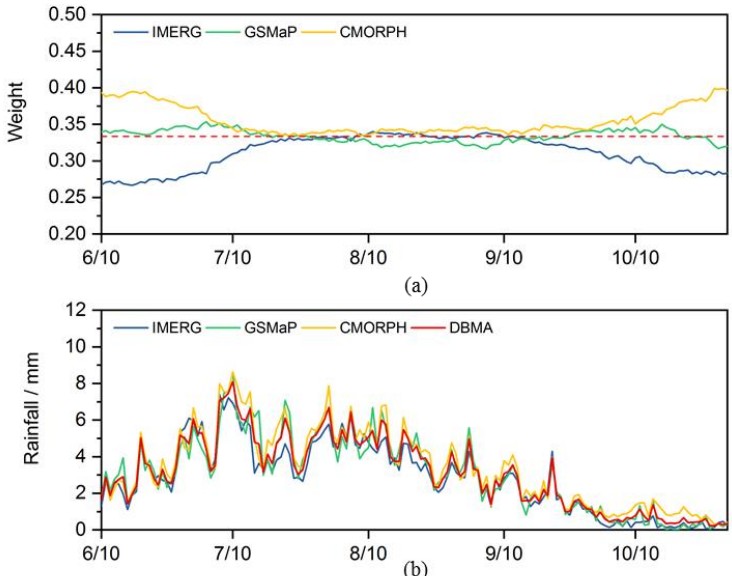


Figure 4. Seasonal variations in basin-averaged (a) weights and (b) rainfall estimates of the
multiyear daily values of IMERG, GSMaP, CMORPH and DBMA.
**4.2. Statistical Evaluation**
Figure 5 shows the statistical evaluation of the merged and original datasets in the
warm seasons. The statistical indices were calculated for three gauge groups including
the training gauges, the test gauges and all gauges at different elevation bands. The
datasets in general presented comparable performance for the training and test gauge
groups, indicating that the sampling procedure of ground gauges is adequately random.
The comparable performance of merged data in training and test gauge groups
demonstrated robustness of the merging method on varying gauges. In terms of RSME,
CC, and POD, the DBMA-merged data shows much better performance on all gauge
groups and elevation bands than the original satellite datasets. The smallest RSME of
merged data indicate that the total rainfall amount of the merged data during the
evaluation period showed the lowest difference from the total amount of gauged rainfall.
The highest CC and POD highlight the best consistency between merged data and



ground gauge data on days when most regions in the basin were rainy. The RB of
DBMA-merged data is at an intermediate level among the satellite datasets as it is the
weighted average of those three datasets. The higher FAR and lower CSI of DBMA-
merged data could be attributed to that the merging method detected rainfall events
when rainfall estimate is higher than zero in any one of the three satellite datasets and
thus resulted in overestimated rainfall occurrence. The overestimated rainfall
occurrence might have small effects on the estimation of rainfall amount, as most of
the falsely alarmed events were tiny. It is noteworthy that the performance of the
merged data shows smaller variance across elevation bands than that of the original
satellite datasets. This is most likely benefiting from the spatially dynamic optimal
weights for the original satellite data. However, the merged data presented the largest
difference from gauged data at the altitudes of 3000-3500 m, because there are much
less gauges on this elevation zone.

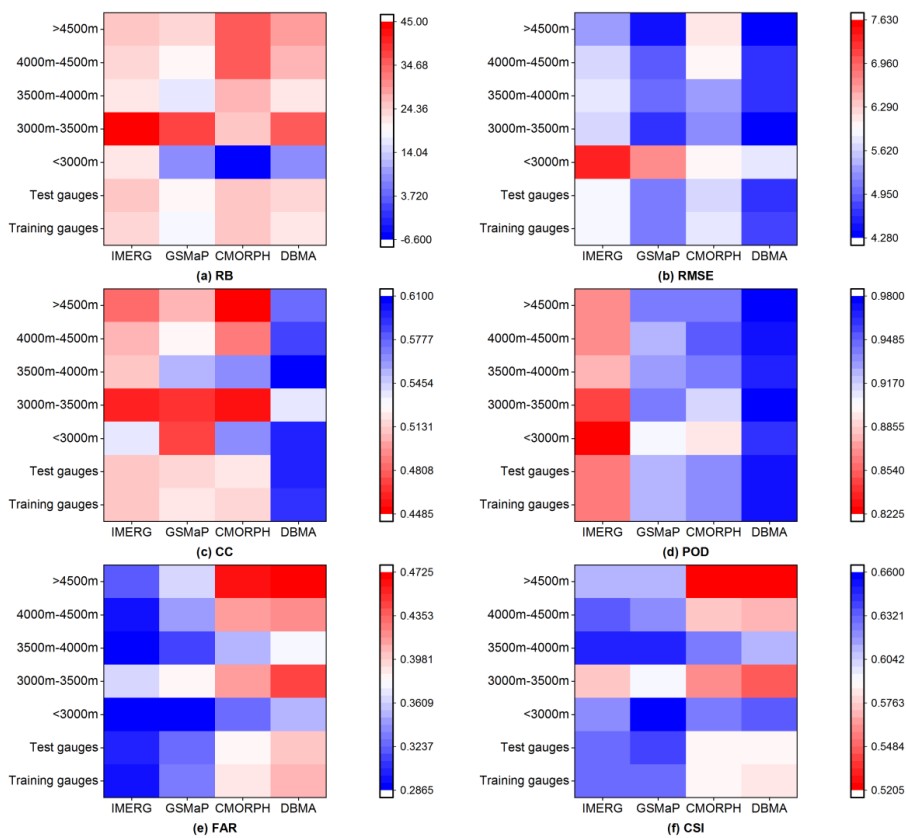


Figure 5. Comparisons of the statistical indices of (a) RB, (b)RMSE, (c) CC, (d) POD, (e) FAR

and (f) CSI for training gauges, test gauges and all gauges at five elevation bands.

Figure 6 shows CC of different datasets on specific gauges. The merged data
presents higher CC values in regions where are densely gauged, i.e., the middle reaches
of YZRB and the east part of the study region, which can be expected as the dense
ground gauges provided strongly informative benchmark likelihoods for the estimation
of satellite data weights. On most of the gauges (Figure 6a), the merged-data presented
higher CC values than the IMERG data, which is consistent with Figure 5c. On contrary,
the merged-data showed reduced CC than GSMaP and CMORPH on more gauges
(Figures 6b-c), indicating that involving IMERG data in the merging procedure on these
gauges lead to deteriorated consistence performance.

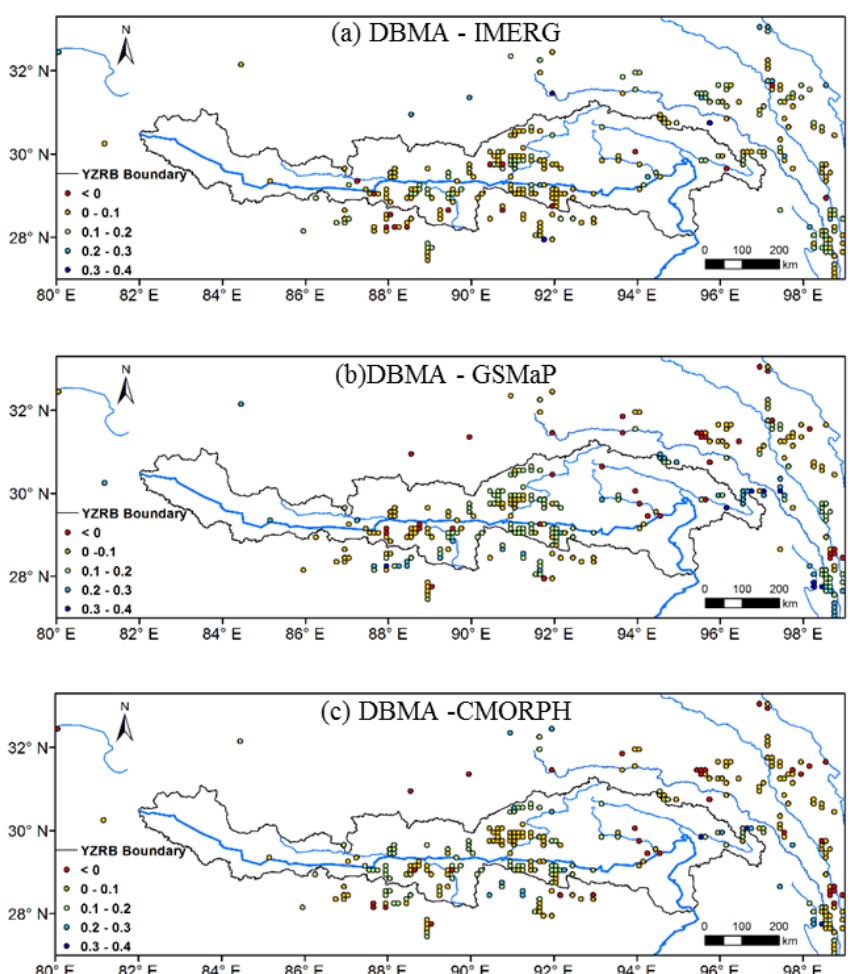

Figure 6. Spatial distributions of CC difference between (a) DBMA and IMERG, (b) DBMA and

GSMaP, (c) DBMA and CMORPH

**4.3. Hydrological Evaluation**

(a) Hydrological simulation

Performance of the THREW model forced by different rainfall datasets are

compared in Table 4. The DBMA-merged dataset achieved the best runoff simulation

among all rainfall inputs, with NSE reaching 0.93 and 0.86 in calibration and validation

period, respectively, indicating an excellent agreement between simulated and observed





hydrographs. Both IMERG and GSMaP underestimated the measured daily discharge,
but the DBMA-merged dataset improved such underestimations (see *RB* values in Table

4).

Table 4. Evaluation metrics of hydrological simulations forced by IMERG, GSMaP, CMORPH
and DBMA.

| Parameters | IMERG | GSMaP | CMORPH | DBMA |
|---|---|---|---|---|
| *NSEcal* | 0.91 | 0.90 | 0.90 | 0.93 |
| *NSEval* | 0.75 | 0.57 | 0.81 | 0.86 |
| *RB* | -0.07 | -0.10 | 0.02 | -0.05 |

(b) Uncertainty analysis
The automatic algorithm pySOT was ran 200 times to investigate the modelling
uncertainty caused by parameter calibration. Figure 7 presents the distributions of *NSE*
values estimated by the ensemble parameter sets of the merged and original rainfall
forces. It is shown that streamflow simulated by the DBMA data at the Nuxia station
presented higher NSEs and smaller uncertainty ranges than that simulated by the
original satellite datasets, indicating that streamflow simulations driven by the merged
dataset showed stronger robustness and were less affected by uncertainty of parameter
calibration.
In addition to the Nuxia hydrological station, model performance on simulating
streamflow at the interior hydrological stations of Yangcun, Nugesha, Gongbujiangda
and Lhasa (Figure 1) were evaluated in Figure 7. It shows that the IMERG forced
simulations presented poor NSE outliers lower than zero at the Lhasa station, in spite
of their good performance at the Yangcun and Nugesha stations; the GSMaP forced
simulations presented large uncertainty ranges in calibration period at Nugesha and
Lhasa, and in validation period at Nuxia and Gongbujiangda; the CMORPH forced
simulations showed the worst performance in validation period at the interior
hydrological stations, despite their sound good performance in calibration period at
Yangcun and Nugesha. In comparison to the satellite datasets, the DBMA forced
simulations tend to perform consistently better with smaller uncertainties at all the





hydrological stations, which can be attributed to that the merged data incorporated the
advantages of different datasets in different regions and temporal periods and thus
better captured the spatial variability of rainfall inputs in sub-basins.

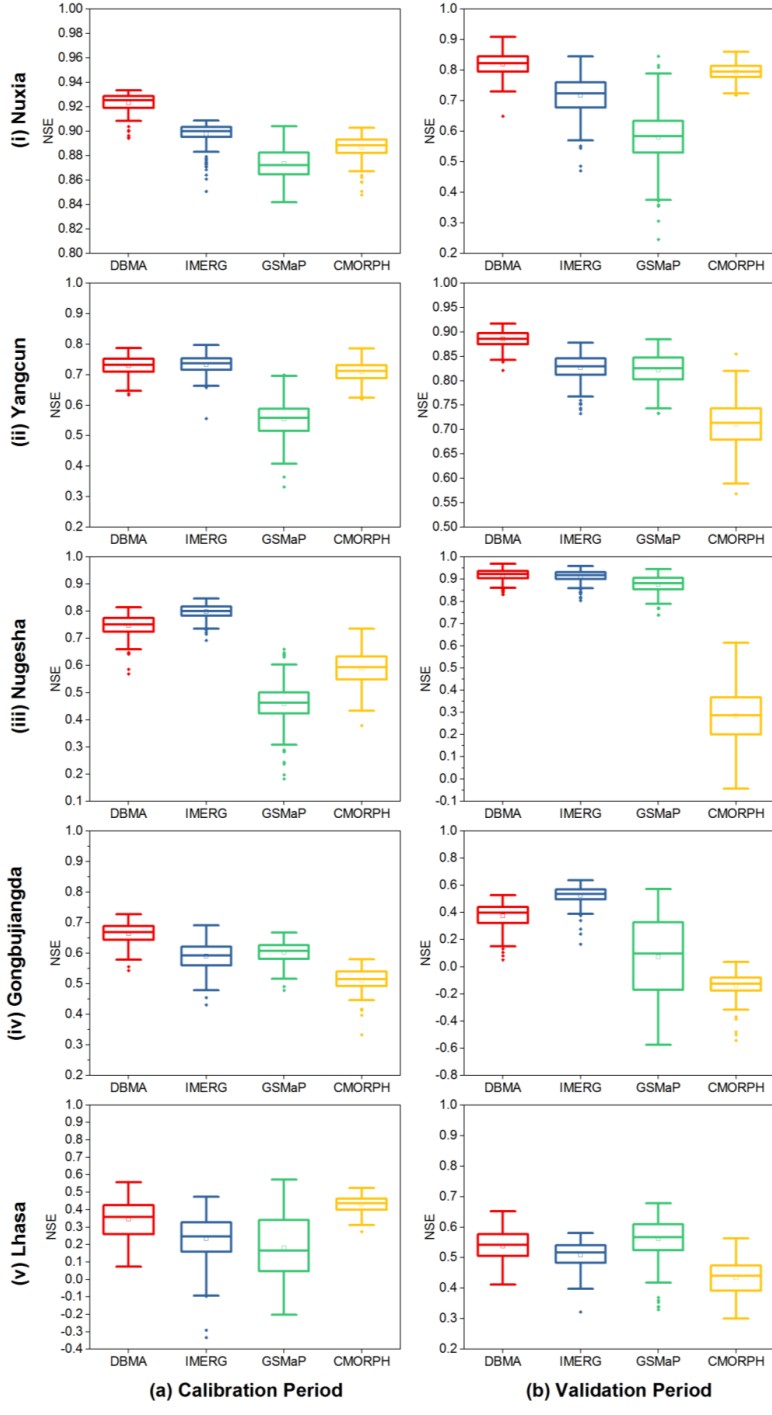


Figure 7. Runoff simulations at Nuxia, Yangcun, Nugesha, Gongbujiangda and Lhasa stations

forced by multiple rainfall inputs.



### 4.4. Comparisons with other datasets

To avoid interference of ground gauge data that merged in the DBMA dataset, the ETC method introduced in Section 3.2 was applied to compare the three merged datasets in Table 5. The RMSE and CC of DBMA calculated by ETC were 1.11 and 0.80, respectively, both of which are obviously superior compared to the corresponding values estimated by CHIRPS and MSWEP, indicating that DBMA data is closer to the true value of rainfall in the study region.

Table 5. Statistical RMSE and CC of merged datasets calculated by the ETC method.

| Datasets | DBMA | CHIRPS | MSWEP |
|----------|------|--------|-------|
| RMSE-ETC | 1.11 | 7.15 | 2.82 |
| CC-ETC | 0.80 | 0.28 | 0.62 |

Runoff simulations forced by the three merged datasets during June 10th 2014 to October 31st 2019 estimated by the corresponding optimal parameter sets were presented in Figure 8. Note that the daily runoff is normalized as Eq. 8 for data security reasons. Simulation by the CHIRPS data presented the lowest performance with NSE values of 0.75 and 0.78 in the calibration and validation periods, respectively. The DBMA forced simulation showed the highest performance with NSE values of 0.93 and 0.86 in the calibration and validation periods, followed by the MSWEP forced simulation which estimated NSE values of 0.9 in the calibration period and 0.76 in the validation period. The performance of streamflow forced by the merged datasets are consistent with the agreements between the merged rainfall estimates and ground truth shown in Table 5.

$$Q_{Normalized}^n = \frac{Q_{sim}^n - \min(Q_{obs})}{\max(Q_{obs}) - min(Q_{obs})} \tag{8}$$

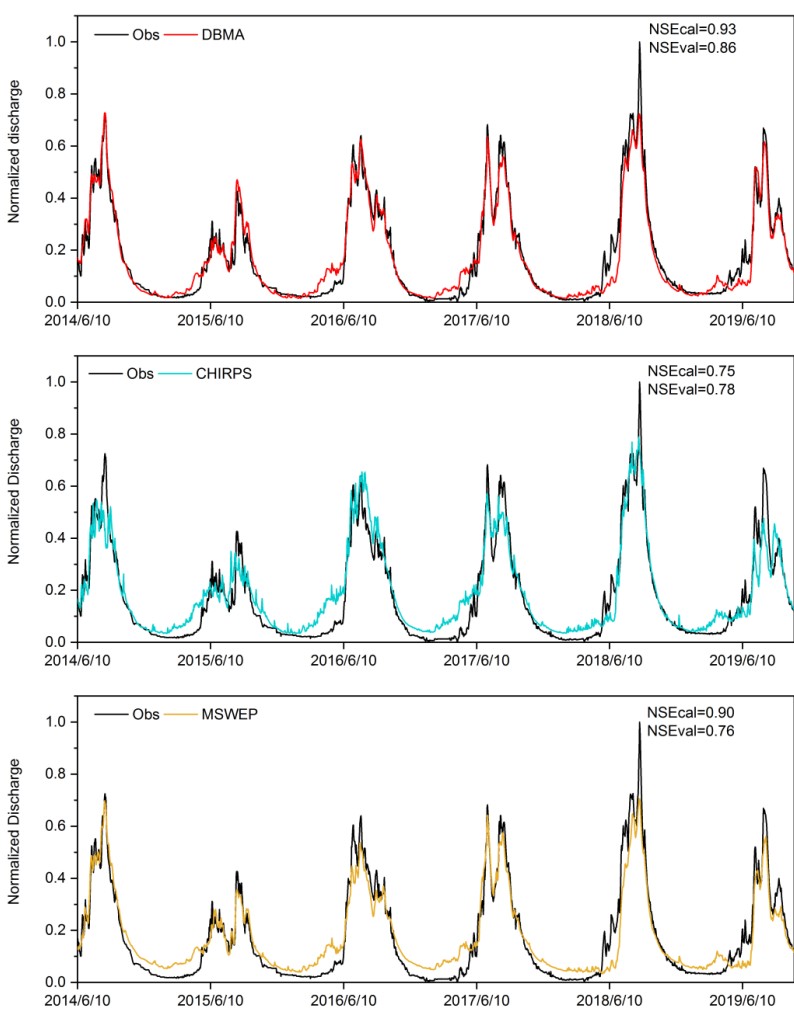


Figure 8. Simulated daily runoff at Nuxia station forced by DBMA, CHIRPS, and MSWEP.
**5 Data Availability**

The high-accuracy rain dataset by merging multi-satellite and dense ground

gauges over southern Tibetan Plateau for the warm seasons in 2014-2019 is freely
accessible at the National Tibetan Plateau Data Center
https://doi.org/10.11888/Hydro.tpdc.271303 (Li et al.,2021).



## 6. Summary

We collated ground-based rainfall observations from a dense gauge network over southern TP. The gauged data provides crucial ground references of measured rainfall. Based on this rain gauge network and three satellite rainfall datasets of IMERG, GSMaP, and CMORPH, a merged rainfall dataset in six warm seasons from June 10$^{th}$ to October 31$^{st}$ during 2014-2019 over the southern TP was established. The DBMA method was used to estimate weights varying in space and time of the three satellite datasets for the merged data. The merged rainfall dataset presented improved performance on representing the total amount of rainfall and detecting the occurrence of gauged rainfall events, and provide a more reliable forcing for hydrological simulations in the YZRB, compared to the original satellite datasets. Comparisons with previous merged rainfall datasets of CHIRPS v2.0 and MSWEP v2 that used relatively sparse rain gauges in the study area demonstrated high values of the newly installed rain gauges for providing robust ground reference for the merging of current satellite datasets. Our results indicated that the merged datasets can meet the critical needs of accurate forcing inputs for the simulations of warm season floods and the robustness calibration of hydrological models.

## Author contribution

TF and LK designed the research. LK, XR, MY developed the approach and datasets. LK downloaded the datasets and performed most of the computation and analysis work. YL, HZ, LH and MY contributed to the revising of the paper.

## Competing interests

The authors declare that they have no conflict of interest.



**Acknowledgements**
Ground gauge data from the hydrological bureau of MWR is acknowledged here.
**Financial support**
This research has been supported by the National Natural Science Foundation of
China (grant no. 92047301, 51825902, 51961125204).

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
