# Peer review of "and Dense Gauges over Southern Tibetan Plateau for 2014-2019"

_Earth System Science Data, 2021_

## Author Response (AR2)

**Authors' response to Reviewers (manuscript: essd-2021-179)**

Dear Editor,

Thanks for your revision and valuable suggestion. We have revised the manuscript about the reasons on limiting the range of the data set. Please see the line of 128-132 in the manuscript. Below are responses to comments of reviewers.

Yours Sincerely,
Kunbiao Li

**Referee #1**

General Comments:

The presented work aims to provide a high-accuracy rainfall dataset by merging all available ground precipitation gauges and three good-quality satellite precipitation datasets over the southern Tibetan Plateau for the warm seasons from 2014 to 2019. The presented results indicate that the merged datasets can actually meet the critical needs of accurate forcing inputs for the simulations of warm season floods and the robustness calibration of hydrological models.

The manuscript is original and is of high scientific quality and significance. The material is very well presented.

The overall scientific impact of this work could be further increased by highlighting more clearly which wider and interdisciplinary relevant implications the presented findings can have beyond the described simulations of warm season floods and the robustness calibration of hydrological models. For example, how can ongoing and future research on fluvial sediment transport, which is very closely related to the use of hydrological models and which has experienced increasing attention during the past years in this study region, benefit from the highly valuable results presented here?

Altogether, this is excellent work which should be used by the interdisciplinary scientific community in the most efficient way possible.

Response: Thanks for your good words and valuable suggestions. We agree that it is important for us to highlight wider and interdisciplinary implications of this work.

Change: According to your suggestions, one paragraph has been added in the revised version. Please see the line of 391-395 in the manuscript.

**Referee #2**

General Comments:
The manuscript presents a daily rainfall dataset merged from three different satellite products and ground rain gauge data from the southern Tibetan Plateau for the warm season. The manuscript is well written and provides a relevant high-quality data set. I only have few minor comments.
Response: Thank you for the kind words. The referee's comments are addressed on a point-to-point basis in the following.

Specific comments:
1) Limitation to warm season. Please explain the reasons for limiting the data set to the range of June 10 to October 31. What would be needed for a year round product?
Response: Thanks for your comments. The range of this dataset is limited to the gauge data. The gauge data used in our study was through the strict quality control methods. In cold seasons there are many missing values and only few gauges meet the requirements. So the warm seasons were selected as the study period to maintain the high quality. While rainfall gauged data is continuously collected to update our merged rainfall data.
Change: The discussion has been added in the manuscript (line 128-132).

2)Why does the number of rain gauges vary over the different years? Please add an explanation. Line 74 mentions 440 new rain gauges, but Figure 1 shows less than 400 in each year. Is 440 the number of different gauge locations? Please specify.
Response: Thanks for pointing it out. The origin ground rainfall data at different gauges has varied quality. In each year, the quality control methods were conducted independently, thus the number of gauges differs in each year. The number of different gauges totally involved in 6 years is 440, shown in Figure 1(c) as black dots. Figure S1 shows the location of gauges in each year, and the number of gauges in Figure S1 is listed in Table 1.
Change: The explanations have been added in the manuscript (Line 75).

3)Training data set: Lines 157-159 state that the training period is 40 days. Which 40 days were used? Then later (lines 171-173) mention that the training set consisted of 85% of the gauges. Is this the same training data set? Please make this clearer.
Response: Thanks. The training period is 40 days before the day, when rainfall data is going to be merged. For example, the weights on June 10th are calculated by the rainfall data from May 1st to June 9th (40 days). It is necessary for Dynamic Bayesian Model Averaging method to obtain the Bayesian weights. For independence, we randomly separated 85% of gauges as training dataset and 15% as test dataset used to verify the merged results. This is the same training dataset.
Change: These explanations have been added in the manuscript (Line 178).

4)Data Availability: Is the underlying rain gauge data as well as the streamflow data publicly available?

Response: The rain gauge data and the streamflow data have not been publicly available because of confidential reasons. Sorry about that.

5)Lines 97-98 in the manuscript state: "Tibetan Plateau, known as the Asian water tower, borders India, Myanmar, Bhutan and Nepal to the south and Pakistan to the west." To me this sounds like these countries only border the Tibetan Plateau, but not that the Tibetan Plateau covers part of these countries. However, Wikipedia states: "The Tibetan Plateau […] is a vast elevated plateau in Central Asia, South Asia and East Asia, covering most of the Tibet Autonomous Region, most of Qinghai, Northwestern Yunnan, Western half of Sichuan, Southern Gansu provinces in Western China, the Indian regions of Ladakh and Lahaul and Spiti (Himachal Pradesh) as well as Bhutan. Response: Thanks for this comment. We agree that this expression is not clear.

Change: To avoid ambiguity, this sentence has been changed as "Tibetan Plateau, known as the Asian water tower, mainly covers parts of China, India, Myanmar, Bhutan, Nepal and Pakistan." (Line 97-98).